# Matting Algorithm with Improved Portrait Details for Images with Complex Backgrounds

Rui Li [1,2,3,4,*], Dan Zhang [1,4,*], Sheng-Ling Geng [1,3,4] and Ming-Quan Zhou [1,3,4]

1   School of Computer Science, Qinghai Normal University, Xining 810000, China; geng_sl@126.com (S.-L.G.); mqzhou@bnu.edu.cn (M.-Q.Z.)
2   School of Computer and Software, Nanyang Institute of Technology, Nanyang 473000, China
3   Academy of Plateau Science and Sustainability, People's Government of Qinghai Province & Beijing Normal University, Haihu, Xining 810004, China
4   The State Key Laboratory of Tibetan Intelligent Information Processing and Application, Qinghai Normal University, Hutai, Xining 810008, China
*   Correspondence: 3162067@nyist.edu.cn (R.L.); danz@mail.bnu.edu.cn (D.Z.)

**Abstract:** With the continuous development of virtual reality, digital image applications, the required complex scene video proliferates. For this reason, portrait matting has become a popular topic. In this paper, a new matting algorithm with improved portrait details for images with complex backgrounds (MORLIPO) is proposed. This work combines the background restoration module (BRM) and the fine-grained matting module (FGMatting) to achieve high-detail matting for images with complex backgrounds. We recover the background by inputting a single image or video, which serves as a priori and aids in generating a more accurate alpha matte. The main framework uses the image matting model MODNet, the MobileNetV2 lightweight network, and the background restoration module, which can both preserve the background information of the current image and provide a more accurate prediction of the alpha matte of the current frame for the video image. It also provides the background prior of the previous frame to predict the alpha matte of the current frame more accurately. The fine-grained matting module is designed to extract fine-grained details of the foreground and retain the features, while combining with the semantic module to achieve more accurate matting. Our design allows training on a single NVIDIA 3090 GPU in an end-to-end manner and experiments on publicly available data sets. Experimental validation shows that our method performs well on both visual effects and objective evaluation metrics.

**Keywords:** portrait matting; background restoration; fine-grained matting; a priori modeling; alpha matting

## 1. Introduction

The purpose of portrait matting is to obtain the foreground of the desired portrait from the input image or video and effectively remove the background information and, at the same time, avoid the loss of portrait detail and avoid artefacts for images in motion or with an auxiliary background. It is one of the key techniques in image processing and has a wide range of applications in practical scenarios, such as in image and video editing [1]. Image matting refers to extracting precise alpha matte from natural images [2]. In previous work, various portrait matting methods [3–5] have achieved impressive results, but there are still great challenges in acquiring details such as portrait hair, glass, and other details, as well as matting complex backgrounds, motion blur, and transparency changes.

Regarding image matting methods, many algorithms have been developed. The background matting template proposed by Soumyadip [6] and others has attracted the attention of a wide range of researchers with its cool technique, which has good results in matting details such as hair, glass, and translucent objects. The limitation of this method is that it requires the input of an image with the same background to help accomplish

alpha matte in the uncertain region. Subsequently, Lin et al. [7] proposed the background matting V2(BGMv2), which focuses on high-resolution video matting with improved accuracy and efficiency but needs help with the auxiliary input and is still unsatisfactory in dealing with large dynamic video matting. Zou [8] introduced sparse and low-rank representations to construct nonlocal structures, producing better matting results regarding spatial and temporal consistency. Sun [9] proposed a deep learning-based matting framework that employs a novel and effective spatiotemporal feature aggregation module (ST-FAM). A lightweight interactive trimap propagation network achieves good results. Kong et al. [10] proposed a semi-supervised deep learning matting algorithm based on semantic consistency of trimaps. The trimap-based approach has good results and reduces the difficulty of the problem, but making the trimap is very costly and suffers in real-world applications. Song [11] proposed a new trimap-free video matting method based on an attention mechanism. In 2022, Lin [12] proposed the robust high-resolution video matting (RVM) method, using a looping architecture and utilizing the temporal information in the video, achieving significant improvements in temporal consistency and matting quality. The method no longer requires any auxiliary input, and its limitation is that it will affect the matting accuracy when the background is too complex or when there are nontargets other than the target characters appearing in the background. The handling of complex backgrounds is still a significant problem to be solved in the field of video matting. In addition to this, the treatment of dynamic backgrounds is also an important topic. Ke et al. [13] proposed a lightweight matting objective decomposition network (MODNet) for portrait matting in real time with a single input image, with no need for additional inputs. Sun et al. [14] proposed MODNet-v on the basis of MODNet. Its architecture is based on the observation that the background of a video frame can be restored by accumulating the background information from historical frames. Chen et al. [15] purposed PP-Matting, a trimap-free architecture that can achieve high-accuracy natural image matting.

In addition, many semantic segmentation-based models have been proposed to implement image and video matting. However, these models may be more effective when dealing with complex scenes and moving objects. The background has different opacity, rigidity, texture, shape, and motion [16]. Currently, we need to solve the problem of detailed matting of hair, glass, and translucent objects for complex scenes. Therefore, a lightweight model that can handle complex background images and provide details of the captured target is needed to provide real-time live video matting on resource-limited platforms such as mobile devices and browsers. In order to better utilize the details and the lightweight framework for complex backgrounds, we propose to apply a background restoration approach combined with the addition of a fine-grained matting module to address the problem of achieving fast matting in complex scenes. Figure 1 shows a comparison of the matting.

In this paper, we perform two main tasks. First, we perform a background restoration where we observe that by dynamically accumulating and updating the background content of consecutive video frames, we can recover meaningful background images whose content changes dynamically with the video background, thus providing an a priori basis for the matting task. We employ a background restoration module to help dynamically restore the image background. Second, fine-grained detail matting is used to solve problems such as hair and detail blurring during the matting process. It uses the background features of the current frame as a priori for the matting model, maintains a high resolution while extracting features at different levels instead of using a downsampling–upsampling codec structure to generate predictive alpha matting, and extracts fine-grained details in the foreground while keeping the feature resolution constant. Finally, background restoration is used as a priori to help preserve details and generate more accurate alpha matte.

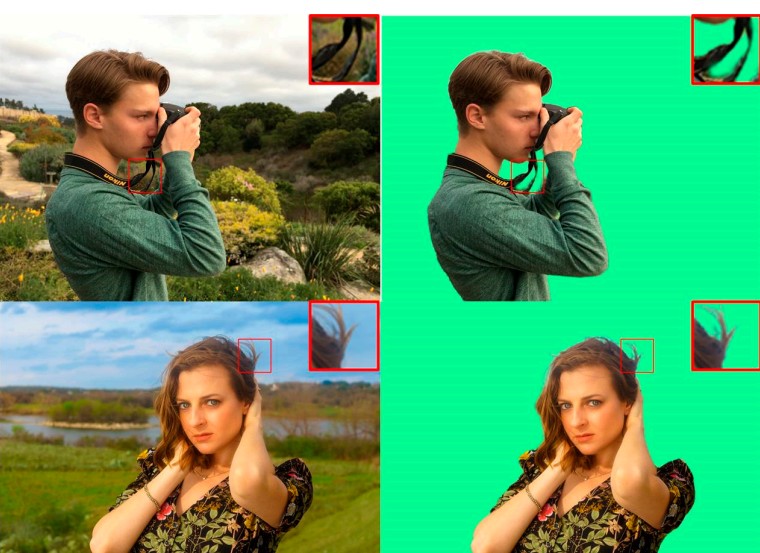

**Figure 1.** Comparison of matting: (**left**) is the original image, (**right**) is the matting results. The red box shows the detail of the matting.

We use background restoration as a matting prior and combine BRM with fine-grained matting to form a new image matting model called MORLIPO. Thanks to the background prior and fine-grained matting provided by MORLIPO, the model has one-third fewer parameters and better performance compared to MODNet. Our new model greatly improves model size, inference speed, and matching performance/stability.

The main contributions of this paper are as follows:

1.  We propose a method that combines background restoration and acquisition of fine-grained matting to simultaneously improve the performance of image matting from two aspects.
2.  We incorporate an ultralightweight model for image matting in complex backgrounds that achieves an optimal trade-off between performance and inference speed. Extensive evaluation of the data set demonstrates the superiority of our model.

The model applies to a variety of complex scene matting, character appearance matting and video conference action background replacement. This study provides an in-depth discussion of the issues in image and real-time video matching to facilitate further research in this area.

The rest of the paper is organized as follows. Section 2 describes the existing image matching methods in detail. Section 3 briefly describes the MORLIPO matting proposed in this paper. In Section 4, we verify the effectiveness of the proposed method through comparative experiments. Finally, in Section 5, we discuss the limitations and significant challenges of the method and present an outlook.

## 2. Related Work

Image matting refers to the precise extraction of the soft matte from foreground objects in arbitrary images [1]. It is a key technology in image editing and film production and effective natural image matting methods can greatly improve current professional workflows. According to the target, matting can be divided into natural matting and portrait matting [2]. The common techniques are image matting and video matting. It can also be classified by whether or not it uses auxiliary input. Methods with auxiliary input generally include trimap, sparse graffiti, background images, user clicks, and so on. Methods without auxiliary input can automatically extract the foreground image or automatically restore the background image [5]. Image matting methods without auxiliary

input are more adaptable to practical needs. In previous solutions, the matting problem is usually transformed into the following formula for ease of understanding [12]:

$$I = \alpha F + (1 - \alpha)B \tag{1}$$

where $F$ is the prospect, $B$ is the background, and $\alpha$ is the transparency, which can also be expressed as the probability of indication that the pixel is in the foreground. The image $I$ represents a linear combination of foreground $F$ and background $B$ controlled by the coefficients. It can also be expressed as

$$I' = \alpha F + (1 - \alpha)B' \tag{2}$$

where $B'$ denotes the target background and image $I'$ indicates the replacement of the original background $B$, using $B'$ to compose the new image [11].

### 2.1. Image Matting

Most existing matting methods exploit low-level properties of image pixels with an auxiliary input as a priori. On the contrary, recent deep learning-based methods utilize high-level semantic information from neural networks to improve the matting results significantly. For example, Cho et al. [17] proposed to combine neural networks with tight form matting [18], KNN matting [19] and KNN segmentation [20]. Xu et al. [21] introduced the first neural network for end-to-end image matting. These methods have different inputs: natural and portrait images with varying categories of objects. Yang et al. [22] developed a multicriteria matting algorithm via the Gaussian process, which searches for the optimal pixel pair by using the Gaussian process fitting model instead of solving the original pixel pair objective function. Qiao et al. [23] used an attentional mechanism for natural images with different object classes, and Zhang et al. [24] combined foreground and background likelihood maps for alpha prediction. For matting portrait images, Shen et al. [25] used a complete convolutional network to generate pseudo-triangular maps before learning the Laplace matrix using an image matting layer. Zhang et al. [26] designed a semi-supervised network to reduce complete dependency on labeled data sets. Chen et al. [19] first predicted low-resolution segmentation maps, which were then used as matting guides. Sengupta et al. [6] provided an additional background image as an alternative auxiliary cue to predict the alpha matte and foreground. Lin et al. [12] had a good performance for high-resolution image matting but also clarified that the method makes it difficult to produce accurate matting with complex backgrounds. In summary, precise matting suitable for complex backgrounds is necessary.

### 2.2. Video Matting

We can perform video matting based on image matting, but it can lead to inefficiency, poor matting quality, time uncorrelation, and other issues. Lin et al. [7] also provided an additional background image for real-time portrait video matting. This extra background is a vital prerequisite for improving video matting performance, as demonstrated by Lin et al. However, two significant drawbacks exist to using a captured background image as a priori, dramatically limiting the application of background matting in practice. First, a captured background image is used. In that case, the background must be static, which means that even small disturbing ground behind it (e.g., light variations, slight jitters, and complex backgrounds) will affect the matting results. Second, the background image requires extra effort from the user to capture, and this process must be performed carefully to ensure that the obtained background image is aligned with the video sequence. Previous trimap-based video matting methods consider the frames' temporal relationship and generate coherent trimaps or matting by performing spatiotemporal optimization of the video sequence. Li et al. [27] proposed video instance matting (VIM), that is, estimating alpha mattes of each instance at each frame of a video sequence. It incorporates temporal mask and temporal feature guidance to improve the temporal consistency of alpha matte

predictions. Some techniques [2] encode temporal coherence using nonlocal extinction Laplacian operators over multiple frames. Recently, video matting methods without trimap have attracted much attention. Elcottet et al. [28] introduced a trimap-free, high-quality neural matte extraction approach that specifically targets the assumptions of visual effects production. Ke et al. [29] divided the matting target into three subtargets and learned the consistency of each subtarget in real-world portrait matting. However, the method needs interframe prediction consistency due to the lack of comprehensive consideration of temporal relationships. To eliminate the above problems, in this paper, we use the BRM module that recovers the background from historical frames, obtains the prior from current frames, and then uses it as the matting prior for the current or future frames.

### 2.3. Scene Background Modeling

Scene background modeling allows you to obtain an initial background model describing a scene without foreground objects. The general problem of background initialization is also referred to as bootstrapping, background estimation, background reconstruction, initial background extraction, or background generation [30]. Laugraud proposed a stationary background generation method LaBGen [31]. An essential component of LaBGen is its flexible motion detection mechanism based on interchangeable background subtraction algorithms. The background image is generated by blending the selected intensities with a median filter. Laugraud proposed LaBGen-P-Semantic on this basis to implement semantic segmentation based on context generation [32]. Djerida et al. [33] proposed a robust background generation method, which can estimate the background from frames that all contain foreground. To refine the motion pixels and reconstruct the background, a refinement algorithm is developed to select the frames that can lead to reliable background estimation. Kajo et al. [34] proposed a self-motion-assisted tensor completion method to overcome the limitations of spatiotemporal slice-based singular value decomposition (SS-SVD) in complex video sequences and enhance the visual appearance of the initialized background. With the proposed method, the motion information, extracted from the sparse portion of the tensor slices, is incorporated with the low-rank information of SS-SVD to eliminate existing artifacts in the initiated background. The CNN-based methods need help in considering long-term temporal information. Both classical and deep learning-based algorithms use background model images to address this limitation. However, obtaining complex background model images is also a complex problem because most algorithms need to spend much time initializing the background model image. To solve this problem, Kim et al. [35] proposed an algorithm for generating background model images based on a deep learning-based segmenter, which is more effective in recovering the background. Sauvalle et al. [36] proposed a new iterative background reconstruction algorithm that uses the current background estimation to guess which image pixels are background pixels and uses only these pixels for new background estimation [37]. The background reconstruction is mainly performed for complex scenes with illumination changes, intermittent object motion, and high levels of clutter. MODNet-v's BRM module is also available by accumulating background information from the current frame and accumulating it [14]. In this paper, we achieve the matting of complex backgrounds by using the BRM module to reduce the effects of dynamic backgrounds, lighting variations, and other factors.

## 3. Method

To solve the complex background and detail matting problem in image matting, we designed and implemented the MORLIPO network by effectively combining the image background restoration and detail matting modules. Figure 2 shows the network structure of MORLIPO. The network consists of a background restoration module (BRM) and fine-grained matting (FGMatting) for background restoration and detail matting, respectively. To generate the final alpha, the background caused by the background restoration is used as an auxiliary input to facilitate the differentiation of a more apparent foreground, and

the detail matting module enhances each other and finally generates a fused alpha, which better extracts the target details while realizing the image matting of complex backgrounds.

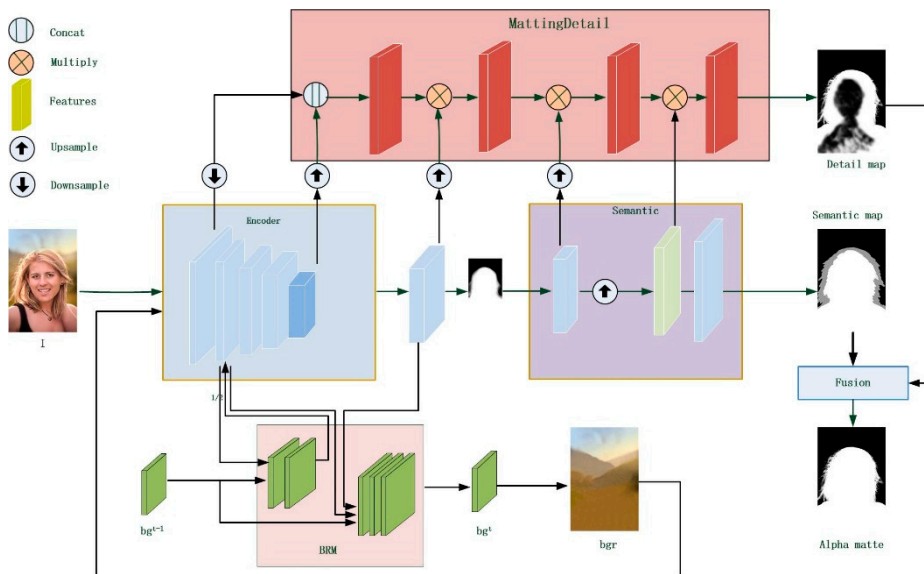

**Figure 2.** MORLIPO architecture. The framework consists of the backbone network, a background recovery module (BRM), and fine-grained matting (FGMatting).

### 3.1. Architecture of the Backbone Network

Inspired by the MODNet [13] approach, our backbone network also uses an image encoder and a semantic fusion module as our backbone network. The approach provides a lightweight image matting model, MORLIPO, which combines the background restoration module BRM and the fine-grained matting module (FGMatting). In our model, the first step is to input only image I. The low-level semantics are extracted by the encoder, which, combined with the background input from the background restoration module of the BGM, makes it easier to perform semantic estimation, because it does not carry a separate decoder, which ensures the validity of semantic estimation and avoids bulky parameters. The subsequent semantic fusion module and fine-grained detail matting are jointly optimized to facilitate the extraction of detailed features such as hair and glass. Meanwhile, to facilitate real-time interaction, we adopt the lightweight model MobileNetV2.

### 3.2. Background Restoration Module

In the BRM framework, we perform background restoration based on image features on its input image I. Background restoration can be used not only for single-image background restoration but also for video image restored. In single-image matting, we directly pass the recovered background bgr back to the backbone network as an input to assist in semantic matting and provide a background prior to predicting that the current image is alpha matte. For background restoration of video images, it is necessary to add a timestamp t, which is used to save the current frame information. The background bgr recovered from the BRM can be used as the background before the next frame. The background restoration module has two important roles: the first role is for recording the background information of the current frame, and the second role is to use the background information of the previous timestamp as a priori for the background information of the current frame. The advantage of this is that as frames accumulate, the acquired prior helps to predict the value of alpha more accurately.

### 3.3. Fine-Grained Detail Matting

As shown in Figure 2, detail matting is usually preserved in high-resolution feature maps. However, existing encoder–decoder segmentation networks with downsample–

upsample structures cannot efficiently maintain high-resolution representations. This work proposes a detail matting module (FGMatting) to obtain fine-grained details. Precisely, FGMatting consists of three residual blocks and a convolutional layer. The initial input to FGMatting combines two intermediate features with corresponding upsampling scales, which contain low-level texture and high-level abstraction information. A semantic context is obtained from the semantic module using a bootstrap flow between the two blocks. The output of FGMatting is a detailed map focusing on the precise representation of the transition regions, which is then fused with the semantic map in the semantic module to generate the final alpha matte.

### 3.4. Loss Function

Our model leverages three losses. The first loss $\mathcal{L}_\alpha$ is the same as MDONet to learn the alpha matte. The second loss $\mathcal{L}_{bg}$ is the explicit constraint to measure the difference between the ground-truth background image and the background image predicted, as

$$\mathcal{L}_{bg} = \sum_{t=1}^{N} \gamma \sqrt{(bg_p^t - bg_g^t)^2 + \varepsilon^2} \tag{3}$$

The $\gamma$ is a binary mask that equals to 4 in the portrait boundaries otherwise 1, and $\varepsilon$ is a small constant value. They can help the loss function to better identify boundaries.

The third loss $\mathcal{L}_f$ is the fusion loss in the final alpha matte

$$\mathcal{L}_f = \sum_{t=1}^{J} \left( \mathcal{L}_\alpha^i(p) + \mathcal{L}_g^i(p) + \mathcal{L}_c^i(p) \right) \tag{4}$$

where $\mathcal{L}_\alpha^i$ is the first loss, $\mathcal{L}_g^i$ is the gradient loss, and $\mathcal{L}_c^i$ is the composition loss. $J$ denotes all pixels in the image.

The final loss $\mathcal{L}$ is calculated as

$$\mathcal{L} = \lambda_1 \mathcal{L}_M + \lambda_2 \mathcal{L}_{bg} + \lambda_3 \mathcal{L}_f \tag{5}$$

For our experiments, we empirically determined these weights to be $\lambda_1 = \lambda_2 = \lambda_3 = 1.0$.

## 4. Experiments

In this paper, the algorithm implemented in the PyTorch 1.1 framework is trained and tested on the Core i7-3770k CPU made by Intel Corporation (Santa Clara, CA, USA) and the TITAN X GPU made by NVIDIA (Santa Clara, CA, USA) platforms to compare the difference between four commonly used image matting algorithms and our proposed algorithm. Four widely used objective evaluation metrics are used to evaluate the experimental results. For the comparative study, we have chosen several classical image matting methods. First, the data set is presented. Second, the methods are compared. Then, the quantization is illustrated and discussed. Finally, the conclusions of this section are drawn.

### 4.1. Data Set

We carried out experiments on four public data sets: alphamatting.com [21], Adobe Composition-1k [38], PPM-100 [13], and DVM [9]. The alphamatting.com data set is a benchmark for existing image matting methods. Due to their small data size, it contains eight test images, which we only use as a test visualization. Composition-1k has 431 foreground images and 50 in the test set. In PPM-100, there are 100 portrait images in the training set with 100 corresponding matte images, which has the advantage of having relatively complete portrait images with related actions that are practical and complex backgrounds that are closer to nature. DVM is a new video matting data set. It consists of real foreground videos, their underlying alpha matting, and background videos of various natural and realistic scenes. The training set consists of each foreground object from 325

images and 75 videos synthesized with 16 randomly selected background videos to generate 6400 videos as the training set. The test set consists of 50 images, and each object from 12 videos is combined with four background videos, thus generating 248 test samples. The training and test sets are not intersected [9].

### 4.2. Methods for Comparsion

In our experiments, four representative image matting methods are selected for comparison, and we can choose them based on the following principles: (1) the method is related to the proposed BRMFGMatting method or can form a better comparative study, (2) the method significantly impacts the field with many citations, (3) the source code of the method can be searched on the GitHub, and (4) it can also be applied to various image video matting.

We selected four methods to compare with ours in the comparative experiment. BGMv2 [7] assists in recovering the alpha and foreground by inputting an additional background and works better for high-resolution images. RVM [12] uses a looping architecture to utilize the video's temporal information, significantly improving temporal consistency and matting quality. No auxiliary inputs, such as trimap or precaptured background images, are required. MODNet [13] uses a lightweight matting objective decomposition network (MODNet) without additional inputs to simultaneously optimize a series of subobjectives with explicit constraints, which uses an efficient spatial pyramid pooling (e-ASPP) module to fuse multiscale features for semantic estimation, improving model efficiency and robustness. PP-Matting [15] is a trimap-free architecture that enables high-precision natural image matting. This method uses high-resolution detail branching (HRDB) and semantic context branching (SCB) for semantic segmentation subtasks to achieve the detail matting task well.

In this section, we report the evaluation results of our proposed model on three data sets: alphamatting.com, the Composition-1k test set, and the PPM-100 data sets. Both quantitative and qualitative results are shown in this section. We evaluate the quantitative results presented based on the sum of absolute differences (*SAD*), mean square error (*MSE*), gradient error (*Grad*) and connectivity error (*Conn*) [13] as follows:

$$SAD = \sum_i |a_i - a_i^*| \tag{6}$$

$$MSE = \frac{1}{n}\sum_i (a_i - a_i^*)^2 \tag{7}$$

$$Grad = \sum(\nabla a_i - \nabla a_i^*)^q \tag{8}$$

$$Conn = \sum_i (\varphi(a_i, W) - \varphi(a_i^*, W)) \tag{9}$$

For all these metrics, lower values indicate better performance.

### 4.3. Analysis of Experimental Results

To better study the different image matting methods, we conducted quantitative experiments to verify their effectiveness. From each data set, a pair of images analyzed more frequently is selected separately, and four different methods are chosen for comparison. The experimental results are shown below. Table 1 lists the objective performance of other matting methods on the same data set. The value indicating the best performance for each metric is shown in bold. Figure 3 shows the experimental results of the selected methods on the same data set.

**Table 1.** Objective evaluation metrics Equations (6)–(9) of the four methods for image matting on the same data set (bold indicates the best-ranked value).

| Method | Backbone | MAD | MSE | Grad | Conn |
|---|---|---|---|---|---|
| BGMv2 | MobileNetV2 | 33.90 | 28.39 | 2.38 | 4.52 |
| MODNet | MobileNetV2 | 7.36 | 2.6 | 1.58 | 0.60 |
| RVM | MobileNetV3 | 6.36 | 1.47 | 1.03 | 0.45 |
| PP-Matting | MobileNetV3 | 5.91 | 1.21 | 0.76 | 0.39 |
| MORLIPO | MobileNetV2 | **5.81** | **1.20** | **0.50** | **0.39** |

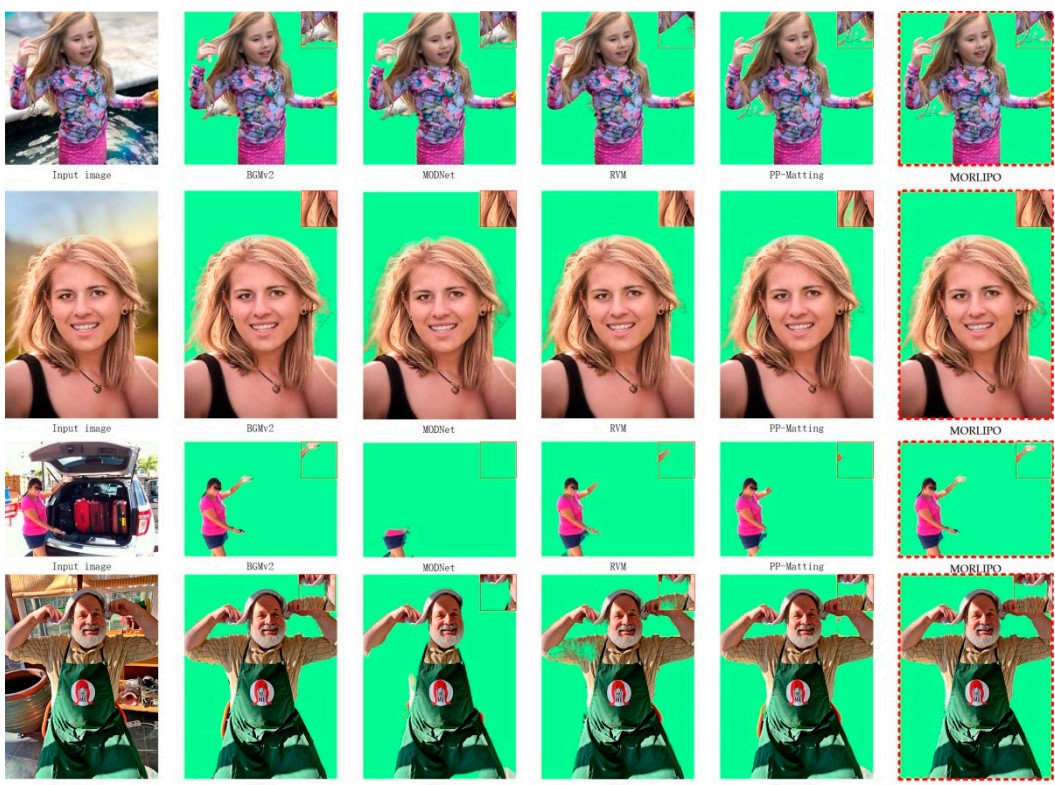

**Figure 3.** Comparison of experiments on the image (the red dashed box indicates better results compared to other methods).

As can be seen in Figure 3, our method performs well in different matting situations, such as hair, holes, large movements, and complex backgrounds. In addition, our method has a clear advantage when dealing with images with complex backgrounds and large movements. Previous methods can only deal with this situation better when there is auxiliary input, such as BGMv2. Still, our method does not need to input an additional background image and can realize fine matting using the background restoration method.

In the video matting, the background is relatively simple, the portrait movement is slight, and several methods are somewhat effective, as shown in Figure 4 in the first and second rows. When the portrait background is relatively complex, multiple target characters appear, as shown in the third row. The background changes quickly, as shown in the fourth row. Our method and the BGMv2 method are visually better. Moreover, our method is easier to apply in real environments by the method of background restoration, which doesn't require additional input.

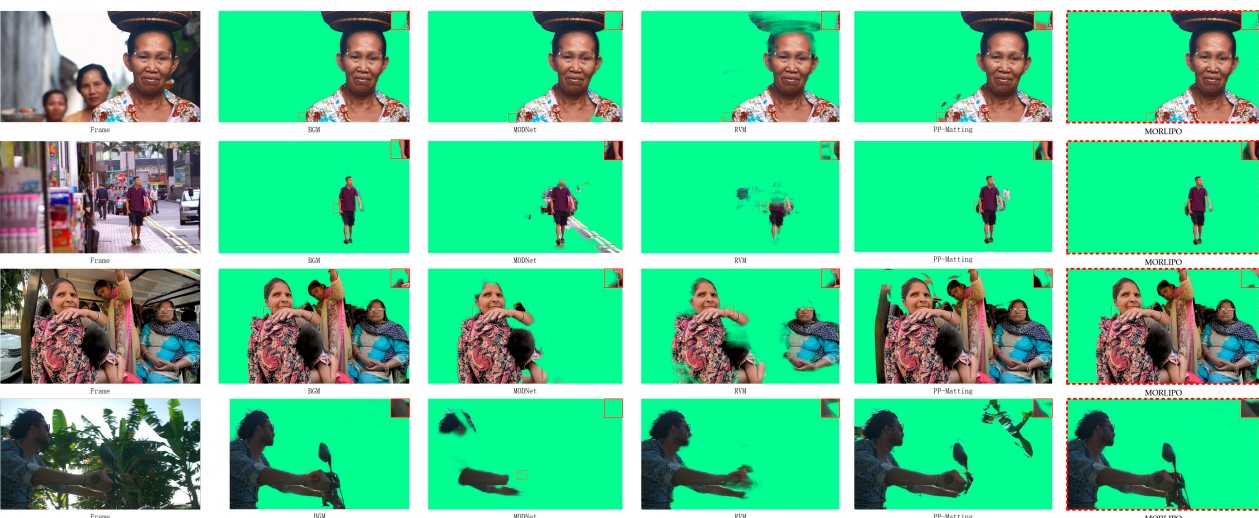

**Figure 4.** Comparison of experiments on the video (the red dashed box indicates better results compared to other methods).

The quantitative evaluation results are shown in Tables 1 and 2, measured by MAD, MSE, Grad, and Conn. Table 1 shows the comparison of the MORLIPO method with the current matting methods for people images. MORLIPO is better than BGMv2, MODNet, RVM, and PP-Matting on all metrics except SAD. Table 2 shows the comparison of the MORLIPO method with the other methods regarding the character matting of the video, and all the indexes show that our method is significantly better than the other methods. All variants of our method give better results than the other methods. The main reason remains that our deep model can understand the complex background of the image and feed the recovered background as an input into the model to help achieve foreground prediction, whereas the other methods cannot. We use MobileNetV2, a model for lightweight mobile devices that is easy to use, has relatively few parameters, and is more efficient. The validation shows that our method performs well on both visual effects and objective evaluation metrics.

**Table 2.** Objective evaluation metrics Equations (6)–(9) of the four methods for video matting on the same data set (bold indicates the best-ranked value).

| Method | Backbone | MAD | MSE | Grad | Conn |
|---|---|---|---|---|---|
| BGMv2 | MobileNetV2 | 31.80 | 30.32 | 32.40 | 5.31 |
| MODNet | MobileNetV2 | 24.04 | 15.53 | 38.88 | 4.28 |
| RVM | MobileNetV3 | 27.50 | 21.31 | 34.18 | 2.12 |
| PP-Matting | MobileNetV3 | 20.03 | 14.32 | 33.45 | 0.49 |
| MORLIPO | MobileNetV2 | **20.01** | **13.49** | **33.40** | **0.47** |

## 5. Conclusions

This paper addresses the problem of portrait image matting for complex backgrounds, designs a method that combines background restoration and detail matting, and verifies the method's effectiveness through experiments. This paper proposes a method, MORLIPO, for portrait image matting of complex backgrounds, which is based on the MobileNetV2 lightweight model. We further construct the background restoration module BRM and add the detail matting module FGMatting to realize the matting of complex backgrounds and details to obtain better matting results. This paper uses a quantitative measure to assess the effectiveness of the proposed method. It conducts experiments on various real image video data sets and compares the performance of MORLIPO with other matting algorithms and previous data. The quantitative measure includes objective metrics such as MAD, MSE, Grad, and Conn indicators to evaluate the quality and stability of the image and

video matting results. Experimental results show that MORLIPO can significantly improve the quality and stability of the image and video matting. The method can be applied to image processing in a wide variety of scenarios involving image editing [39], video conferencing [6], medical imaging [40], cloud detection [41], game production, intelligent transportation [42], and, more recently, multimodal 3D applications [43]. However, our method has limitations for multimodal image editing tasks, and the currently realized multimodal portrait matting is not ideal.

For future work, we plan to apply more advanced multimodal image matting techniques to address the above challenges. First, we will work on creating more efficient data sets for image matting that utilize the powerful learning capabilities of deep learning to obtain better matting results. Second, more complex and powerful feature fusion modules will be added to improve the model's performance. Third, multimodal matting can be further investigated to try to implement portrait matting in conjunction with other modalities such as text, speech, and eye gaze. Hopefully, this study will provide a good reference for researchers studying image matting and give some information about this rapidly developing and important field.

**Author Contributions:** Conceptualization, R.L. and D.Z.; formal analysis, R.L.; methodology, R.L. and D.Z.; validation, R.L.; writing—original draft, R.L.; writing—review and editing, D.Z., M.-Q.Z. and S.-L.G. All authors have read and agreed to the published version of the manuscript.

**Funding:** This work was partially supported by the National Nature Science Foundation of China (No. 62102213, 62262056); Independent project fund of State Key lab of Tibetan Intelligent Information Processing and Application (Coestablished by province and ministry): 2022SKL014; Qinghai Province Key R&D and Transformation Programme (No. 2022-QY-203); National Key R&D plan (No. 2020YFC1523305).

**Data Availability Statement:** The data sets analyzed during the current study are: alphamatting.com: http://www.alphamatting.com (accessed on 23 November 2023); Adobe Composition-1k: https://github.com/PaddlePaddle/PaddleSeg/ (accessed on 23 November 2023); PPM-100: https://github.com/ZHKKKe/PPM/ (accessed on 23 November 2023); DVM: https://pan.baidu.com/s/1yBJr0SqsEjDToVAUb8dSCw Password: l9ck (accessed on 23 November 2023). The contrasting experimental approaches are BGMv2: https://github.com/PeterL1n/BackgroundMattingV2 (accessed on 1 December 2023); RVM: https://github.com/SHI-Labs/VMFormer (accessed on 1 December 2023); MODNet: https://github.com/ZHKKKe/MODNet (accessed on 2 December 2023); PP-Matting: https://github.com/PaddlePaddle/ (accessed on 23 December 2023).

**Acknowledgments:** The authors would like to express their gratitude to the editors and anonymous reviewers for their comments and suggestions.

**Conflicts of Interest:** We declare that we do not have any commercial or associative interests that represent a conflict of interest in connection with the work submitted.

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
