# Peer review of "Matting Algorithm with Improved Portrait Details for Images with Complex Backgrounds"

_applsci, doi:10.3390/app14051942_

Round 1

Reviewer 1 Report

Comments and Suggestions for Authors

This paper proposes a method for portrait image matting of complex backgrounds, which is based on the MobileNetV2 lightweight model. The topic is interesting. The results of the analysis enrich the research in this field. The experimental results show that this method can significantly improve the quality and stability of the video compared with other methods.

However, from my point of view, a number of revisions are needed.

First, the authors must develop the conclusion section adding the impact of this research and clearly mention to whom the obtained results are useful.

Secondly, they should explain their results based on the literature (the bibliography used includes relevant publications in the field)

Overall, I evaluate the study very positively and I recommend its publication after minor revisions.

Author Response

For research article

Response to Reviewer 1 Comments

1. Summary

First, thank you very much for your recognition of our work! It motivates us greatly. Additionally, thank you for your professional comments and suggestions on the manuscript. Based on your professional guidance, we revised this paper according to your comments. Thank you again for your patient guidance!

2. Questions for General Evaluation

Reviewer’s Evaluation

Response and Revisions

Does the introduction provide sufficient background and include all relevant references?

Yes

Are all the cited references relevant to the research?

Yes

Is the research design appropriate?

Yes

Are the methods adequately described?

Yes

Are the results clearly presented?

Yes

Are the conclusions supported by the results?

Must be improved

We have made the appropriate changes and will respond to them in the point-by-point response letter.

3. Point-by-point response to Comments and Suggestions for Authors

Comments 1: First, the authors must develop the conclusion section adding the impact of this research and clearly mention to whom the obtained results are useful.

Response 1: Thank you for pointing this out. We agree with this comment. Therefore, we have summarized the scope of the study's impact. Our manuscript was modified and is marked in yellow in the revised document in page 11.

‘The method can be applied to image processing in a wide variety of scenarios involving image editing [39], videoconferencing[6], medical imaging[40], cloud detection[41], game production, intelligent transportation[42], and more recently, multimodal 3D applications[43].’

Comments 2: Secondly, they should explain their results based on the literature (the bibliography used includes relevant publications in the field)

Response 2:  We agree. We have illustrated the application of image keying in correlated image processing by adding references 39-43. Our manuscript was modified and is marked in yellow in page 13.

39.     Zhang, H.; Zhang, J.; Perazzi, F.; Lin, Z.; Patel, V .M. Deep image compositing. In Proceedings of the IEEE/CVF Winter Conference on Applications of Computer Vision, Waikoloa, HI, USA, 5–9 January 2021; pp. 365–374.

40.     Wang, L.; Ye, X.; Ju, L.; He, W.; Zhang, D.; Wang, X.; Huang, Y.; Feng, W.; Song, K.; Ge, Z. Medical matting: Medical image segmentation with uncertainty from the matting perspective. Comput. Biol. Med. 2023, 158, 106714.

41.     Ma, D.; Wu, R.; Xiao, D.; Sui, B. Cloud Removal from Satellite Images Using a Deep Learning Model with the Cloud-Matting Method. Remote Sens. 2023, 15, 904.

42.     Husain, A.; Maity, T.; Yadav, R. K. Vehicle detection in intelligent transport system under a hazy environment: a survey. IET Image Processing. 2020. 14(1), 1-10.

43.     Maqsood, S.; Damasevicius, R.; SiÅ‚ksa, J.; Woźniak, M. Multimodal image fusion method based on multiscale image matting. In International Conference on Artificial Intelligence and Soft Computing. 2021, pp. 57-68.

4. Response to Comments on the Quality of English Language

Point 1:

Response 1:

5. Additional clarifications

We would like to express our sincere appreciation to the editor and all the reviewers for their valuable comments and suggestions. We revised the paper accordingly. The major issues addressing these comments are explained below and are presented in the revised version of our manuscript. The modified parts are marked in the modified paper. We greatly appreciate your help and that of the reviewers to improve this paper. We hope that the revised manuscript is now suitable for publication.

Reviewer 2 Report

Comments and Suggestions for Authors

The authors should put the URLs of the dataset like a footnote. The authors should apply the same criteria for all the URLs included in the paper.

The validation shows that our method performs well on both visual effects and objective evaluation metrics. The authors should write qualitatively in their results.

The authors should write the results in Table 1  in the results section.

The conclusions should be qualitative, with the following: the paper uses a quantitative measure to assess the effectiveness of the proposed method. It conducts experiments on various real image video datasets and compares the performance of MORLIPO with other matting algorithms and previous data. The quantitative measure likely includes objective metrics such as accuracy, precision, recall, F1-score, or other quantitative indicators to evaluate the quality and stability of the video matting results. 

Comments on the Quality of English Language

The English quality of the provided text is good. The sentences are well-structured and convey the intended meaning effectively. However, an English native should review the article.

Author Response

For research article

Response to Reviewer 2 Comments

1. Summary

First, thank you very much for your recognition of our work! It motivates us greatly. Additionally, thank you for your professional comments and suggestions on the manuscript. Based on your professional guidance, we revised this paper according to your comments. Thank you again for your patient guidance!

2. Questions for General Evaluation

Reviewer’s Evaluation

Response and Revisions

Does the introduction provide sufficient background and include all relevant references?

Can be improved

We have made the appropriate changes and will respond to them in the point-by-point response letter.

Are all the cited references relevant to the research?

Yes

Is the research design appropriate?

Yes

Are the methods adequately described?

Yes

Are the results clearly presented?

Can be improved

We have made the appropriate changes and will respond to them in the point-by-point response letter.

Are the conclusions supported by the results?

Can be improved

We have made the appropriate changes and will respond to them in the point-by-point response letter.

3. Point-by-point response to Comments and Suggestions for Authors

Comments 1: The authors should put the URLs of the dataset like a footnote. The authors should apply the same criteria for all the URLs included in the paper 11.

Response 1: Thank you for pointing this out. We agree with this comment. Therefore, we have standardized the datasets used in the Data Availability Statement and the URLs associated with the comparison experiments have also been collated in page 11. The URL placed in the original article was also removed. Our manuscript was modified and is marked in yellow in the revised document.

Data Availability Statement: The datasets analyzed during the current study are: alphamatting.com: http://www.alphamatting.com; Adobe Composition-1k: https://github.com/PaddlePaddle/PaddleSeg/; PPM-100: https://github.com/ZHKKKe/PPM/; DVM: https://pan.baidu.com/s/1yBJr0SqsEjDToVAUb8dSCw. The contrasting experimental approaches: BGMv2:https://github.com/PeterL1n/BackgroundMattingV2; RVM: https://github.com/SHI-Labs/VMFormer; MODNet:https://github.com/ZHKKKe/MODNet;PP-Matting : https://github.com/ PaddlePaddle/.

Comments 2: The validation shows that our method performs well on both visual effects and objective evaluation metrics. The authors should write qualitatively in their results.

Response 2: We agree. We think your suggestion provides a more accurate description of the results of the comparison experiment. We have made changes in the description of results section in page 11. Our manuscript was modified and is marked in yellow in the revised document.

The validation shows that our method performs well on both visual effects and objective evaluation metrics.

Comments 3: The authors should write the results in Table 1 in the results section.

Response 3: We agree. We think your suggestion describes the results of the experiment more clearly and accurately. We have made changes in the description of results section in page 10. Our manuscript was modified and is marked in yellow in the revised document.

Table 1 shows the comparison of the MORLIPO method with the current matting methods for people images. MORLIPO is better than BGMv2, MODNet, RVM, and PP-Mattin on all metrics except SAD. Table 2 shows the comparison of the MORLIPO method with the other methods regarding the character matting of the video, and all the indexes show that our method is significantly better than the other methods.

Comments 4: The conclusions should be qualitative, with the following: the paper uses a quantitative measure to assess the effectiveness of the proposed method. It conducts experiments on various real image video datasets and compares the performance of MORLIPO with other matting algorithms and previous data. The quantitative measure likely includes objective metrics such as accuracy, precision, recall, F1-score, or other quantitative indicators to evaluate the quality and stability of the video matting results. 

Response 4: Thank you for pointing this out. We agree with this comment. We have made changes in the description of results section in page 11. Our manuscript was modified and is marked in yellow in the revised document.

‘This paper uses a quantitative measure to assess the effectiveness of the proposed method. It conducts experiments on various real image video datasets and compares the performance of MORLIPO with other matting algorithms and previous data. The quantitative measure likely includes objective metrics such as MAD, MSE, Grad and Conn indicators to evaluate the quality and stability of the image and video matting results. Experimental results show that MORLIPO can significantly improve the quality and stability of the image and video matting.’

4. Response to Comments on the Quality of English Language

Point 1: The English quality of the provided text is good. The sentences are well-structured and convey the intended meaning effectively. However, an English native should review the article.

Response 1: Thank you for recognizing our article. Regarding the English writing, we have not only revised it carefully but also asked the English native to review it.

5. Additional clarifications

We would like to express our sincere appreciation to the editor and all the reviewers for their valuable comments and suggestions. We revised the paper accordingly. The major issues addressing these comments are explained below and are presented in the revised version of our manuscript. The modified parts are marked in the modified paper. We greatly appreciate your help and that of the reviewers to improve this paper. We hope that the revised manuscript is now suitable for publication.

Reviewer 3 Report

Comments and Suggestions for Authors

Dear authors,

The aspects that I am going to rate are the following:

  • Novelty. The research question needs to be defined, and all implications of picture matting must be explored. The results do not provide an advancement of the current knowledge.
  • The research results are not appropriately interpreted and there are no stated hypotheses. Further work must be developed.
  • Quality. The article must be rewritten because data and analyses are not presented appropriately.
  • Scientific Soundness. The study needs to be better designed and technically sound. The data analysis is not enough to conclude.
  • Low Interest to the Readers.
  • The study does not advance in the current knowledge. The authors do not address an important long-standing question with smart experiments.
  • Discussions and Limits Sections must be added.
  • The bibliography can be improved.

Success!

Comments on the Quality of English Language

The text requires a few adjustments to the English language with the aim of enhancing its clarity and conciseness

Author Response

For research article

Response to Reviewer 3 Comments

1. Summary

First, thank you for your professional comments and suggestions on the manuscript. Based on your professional guidance, we revised this paper according to your comments. Thank you again for your patient guidance!

2. Questions for General Evaluation

Reviewer’s Evaluation

Response and Revisions

Does the introduction provide sufficient background and include all relevant references?

Must be improved

We have made the appropriate changes and will respond to them in the point-by-point response letter.

Are all the cited references relevant to the research?

Must be improved

We have made the appropriate changes and will respond to them in the point-by-point response letter.

Is the research design appropriate?

Must be improved

We have made the appropriate changes and will respond to them in the point-by-point response letter.

Are the methods adequately described?

Must be improved

We have made the appropriate changes and will respond to them in the point-by-point response letter.

Are the results clearly presented?

Must be improved

We have made the appropriate changes and will respond to them in the point-by-point response letter.

Are the conclusions supported by the results?

Must be improved

We have made the appropriate changes and will respond to them in the point-by-point response letter.

3. Point-by-point response to Comments and Suggestions for Authors

·       Comments 1: Novelty. The research question needs to be defined, and all implications of picture matting must be explored.

Response 1: Thank you very much for your guidance and help with the article, we will elaborate further on the research question and explore the implications of image matting further.

The research question: In page 2.

Currently, we need to solve the problem of detailed matting of hair, glass, and translucent objects for complex scenes.

Image matting: In page 4.

According to the target of matting can be divided into natural matting and portrait matting [2]. The common ones are image matting and video matting. It can also be classified by whether or not using auxiliary input. Methods with auxiliary input generally include trimap, sparse graffiti, background images, user clicks etc. Methods without auxiliary input can automatically extract the foreground image or automatically restore the background image [5].

·       Comments 2:The results do not provide an advancement of the current knowledge. The research results are not appropriately interpreted and there are no stated hypotheses. Further work must be developed.

Response 2: First of all, thank you for your professional advice. We very much regret that we did not manage to elaborate on the results in a detailed and careful manner. For this reason, we have described the results section of the article in more detail. Our manuscript was modified and is marked in yellow in page 10.

·       Comments 3: Quality. The article must be rewritten because data and analyses are not presented appropriately.

Response 3: Thank you for your valuable comments. For more adequate data and analysis, we have added the corresponding analysis in 4.3. Analysis of experimental results. Our manuscript was modified and is marked in yellow in page 10.

4.3. Analysis of experimental results.

The quantitative evaluation results are shown in Table 1 and Table 2, measured by MAD, MSE, Grad and Conn. Table 1 shows the comparison of the MORLIPO method with the current matting methods for people images. MORLIPO is better than BGMv2, MODNet, RVM, and PP-Matting on all metrics except SAD. Table 2 shows the comparison of the MORLIPO method with the other methods regarding the character matting of the video, and all the indexes show that our method is significantly better than the other methods. All variants of our method give better results than other methods. The main reason remains that our deep model can understand the complex background of the image and feed the recovered background as an input into the model to help achieve foreground prediction, whereas the other methods cannot. We use MobileNetV2, a model for lightweight mobile devices that is easy to use, has relatively few parameters, and is more efficient. The validation shows that our method performs well on both visual effects and objective evaluation metrics.

·       Comments 4: Scientific Soundness. The study needs to be better designed and technically sound. The data analysis is not enough to conclude. Low Interest to the Readers.

Response 4: Thank you for your professional input and we have revised our data analysis to support the conclusions reached. Our manuscript was modified and is marked in yellow in page 10.

Table 1 shows the comparison of the MORLIPO method with the current matting methods for people images. MORLIPO is better than BGMv2, MODNet, RVM, and PP-Matting on all metrics except SAD. Table 2 shows the comparison of the MORLIPO method with the other methods regarding the character matting of the video, and all the indexes show that our method is significantly better than the other methods. All variants of our method give better results than other methods. The main reason remains that our deep model can understand the complex background of the image and feed the recovered background as an input into the model to help achieve foreground prediction, whereas the other methods cannot. We use MobileNetV2, a model for lightweight mobile devices that is easy to use, has relatively few parameters, and is more efficient. The validation shows that our method performs well on both visual effects and objective evaluation metrics.

Comments 5: Low Interest to the Readers.

Response 5: Thank you very much for your careful review of our manuscript. We apologize for not engaging our readers as well as we could have. In fact, it was a very interesting and rewarding endeavor. And we are already using it in practical applications. The scope of its application we also further clarified in the conclusion section. Our manuscript was modified and is marked in yellow in page 11.

Comments 6: The study does not advance in the current knowledge. The authors do not address an important long-standing question with smart experiments.

Response 6: Thank you very much for your comment, our method is not particularly innovative, but it does achieve portrait matting through image background recovery and fine-grained matting and gets good results in visual effects and objective evaluation. We will also continue our research in the next work in order to have a bigger breakthrough.

Comments 7:  Discussions and Limits Sections must be added.

Response 7: Thank you for your valuable comments, we then added in the conclusion section the application scenarios of the method for the quantitative analysis of the results of the article. We have also clarified the limitation that since our method does not consider multi-modal portrait matting, it is not possible to realize cooperation with other modal data. In response to the limits, we will also add this aspect of research in our subsequent work, so we have made adjustments to our challenges for the future. Our manuscript was modified and is marked in yellow in page 11.

Discussions: This paper uses a quantitative measure to assess the effectiveness of the proposed method. It conducts experiments on various real image video datasets and compares the performance of MORLIPO with other matting algorithms and previous data. The quantitative measure includes objective metrics such as MAD, MSE, Grad and Conn indicators to evaluate the quality and stability of the image and video matting results. Experimental results show that MORLIPO can significantly improve the quality and stability of the image and video matting. The method can be applied to image processing in a wide variety of scenarios involving image editing [39], video conferencing [6], medical imaging [40], cloud detection [41], game production, intelligent transportation[42], and more recently, multimodal 3D applications[43]. However, our method also has limits, for the multimodal image editing task, with other modalities to achieve the effect of portrait matting not ideal.

Limits: However, our method also has limits, for the multi-modal image editing task, with other modalities to achieve the effect of portrait matting not ideal.

Future work: Third, Multi-modal matting can be further investigated,Try to implement portrait matting in conjunction with other modalities such as text, speech, and eye gaze.

Comments 8: The bibliography can be improved.

Response 8: Thank you for pointing this out. We agree with this comment. We have added the necessary references in the appropriate sections. Our manuscript was modified and is marked in yellow.

1.       Huang, L.; Liu, X.; Wang, X.; Li, J.; Tan, B. Deep Learning Methods in Image Matting: A Survey. Appl. Sci. 2023, 13, 1–22.

2.       Li, J.; Zhang, J.; Tao, D. Deep Image Matting: A Comprehensive Survey. arXiv preprint arXiv:2304.04672. 2023.

10.    Kong, Y.; Li, J.; Hu, L.; Li, X. Semi-Supervised Learning Matting Algorithm Based on Semantic Consistency of Trimaps. Applied Sciences. 2023, 13, 1–17.

22.    Yang, Y.; Gou, H.; Tan, M.; Feng, F.; Liang, Y., Xiang, Y.; Wang, L.; Huang, H. Multi-criterion sampling matting algorithm via gaussian process. Biomimetics. 2023. 8(3), 1-18.

26.  Zhang, X.; Wang, G.; Chen, C.; Dong, H.; Shao, M. Semi-Supervised Portrait Matting via the Collaboration of Teacher–Student Network and Adaptive Strategies. Electronics. 2022,1,1-18.

39.  Zhang, H.; Zhang, J.; Perazzi, F.; Lin, Z.; Patel, V .M. Deep image compositing. In Proceedings of the IEEE/CVF Winter Conference on Applications of Computer Vision, Waikoloa, HI, USA, 5–9 January 2021; pp. 365–374.

40.    Wang, L.; Ye, X.; Ju, L.; He, W.; Zhang, D.; Wang, X.; Huang, Y.; Feng, W.; Song, K.; Ge, Z. Medical matting: Medical image segmentation with uncertainty from the matting perspective. Comput. Biol. Med. 2023, 158, 106714.

41.    Ma, D.; Wu, R.; Xiao, D.; Sui, B. Cloud Removal from Satellite Images Using a Deep Learning Model with the Cloud-Matting Method. Remote Sens. 2023, 15, 904.

42.    Husain, A.; Maity, T.; Yadav, R. K. Vehicle detection in intelligent transport system under a hazy environment: a survey. IET Image Processing. 2020. 14(1), 1-10.

43.    Maqsood, S.; Damasevicius, R.; SiÅ‚ksa, J.; Woźniak, M. Multimodal image fusion method based on multiscale image matting. In International Conference on Artificial Intelligence and Soft Computing. 2021, pp. 57-68.

4. Response to Comments on the Quality of English Language

Point 1: The text requires a few adjustments to the English language with the aim of enhancing its clarity and conciseness.

Response 1: Thank you for carefully reviewing our manuscript. Regarding the English writing, we have not only revised it carefully but also asked the English native to review it.

5. Additional clarifications

We would like to express our sincere appreciation to the editor and all the reviewers for their valuable comments and suggestions. We revised the paper accordingly. The major issues addressing these comments are explained below and are presented in the revised version of our manuscript. The modified parts are marked in the modified paper. We greatly appreciate your help and that of the reviewers to improve this paper. We hope that the revised manuscript is now suitable for publication.

Reviewer 4 Report

Comments and Suggestions for Authors

Matting algorithm with improved portrait details for… , applsci-2752929

1.       Your paper is well written, with minor flaws in the English. It is a valuable report of your work.

2.       You report comparative tests of your results with other methods. Well done!

3.       Your results are good and impressive.

4.       A few papers have not been referred to. The recent papers [1–5] seem related and relevant to this work. You should defend - I think - why you excluded them in your comparison.
For instance: why isn’t the survey by Huan et. al in Applied Sciences
[1] used by you?

Small issues

5.       Line 40, ‘ there have been a large amount algorithms.’ should read ‘many algorithms have been developed.’

6.       Line 45, the shortcut ‘mattingV2(BGMv2)’ should be introduced after mentioning the name of its author. The uninitiated reader is not aware yet what this shortcut is. You might in the beginning refer to the clarifying Tables 1 and 2, wherein all the shortcuts of methods are listed.

7.       Line 97, ‘but better performance’ should read ‘and better performance’.

8.       Line 124, ‘the probability of indicating that the pixel is’ should read ‘the probability indicating that the pixel is’, or should read ‘the probability of indication that the pixel is’.

9.       Line 352, ‘Still, our method is easier to’ should read ‘Moreover, our method is easier to’.

10.   Lines 363, 365, in the capture of the tables,  should have reference – for the quick reader - to the formulas (6) to (9). This could be done as ‘Objective evaluation metrics (6) to (9) of the four methods’.

11.   Line 371, Inconsistency: the name MORLIPO is not in the results Tables 1 and 2.

References

1.        Huang, L.; Liu, X.; Wang, X.; Li, J.; Tan, B. Deep Learning Methods in Image Matting: A Survey. Appl. Sci. 2023, 13, 1–22.

2.        Kong, Y.; Li, J.; Hu, L.; Li, X. Semi-Supervised Learning Matting Algorithm Based on Semantic Consistency of Trimaps. Appl. Sci. 2023, 13, 1–17.

3.        Yang, Y.; Gou, H.; Tan, M.; Feng, F.; Liang, Y.; Xiang, Y.; Wang, L.; Huang, H. Multi-Criterion Sampling Matting Algorithm via Gaussian Process. Biomimetics 2023, 8, 1–18.

4.        Zhang, X.; Wang, G.; Chen, C.; Dong, H.; Shao, M. Semi-Supervised Portrait Matting via the Collaboration of Teacher–Student Network and Adaptive Strategies. Electronics 2022, 11, 1–18.

5.        Zou, D.; Chen, X.; Cao, G.; Wang, X. Unsupervised Video Matting via Sparse and Low-Rank Representation. IEEE Trans. Pattern Anal. Mach. Intell. 2020, 42, 1501–1514.

Comments on the Quality of English Language

Well written, with minor flaws. See my points above: 5, 7, 8 and 9.

Author Response

For research article

Response to Reviewer 4 Comments

1. Summary

First, thank you very much for your recognition of our work! It motivates us greatly. Additionally, thank you for your professional comments and suggestions on the manuscript. Based on your professional guidance, we revised this paper according to your comments. Thank you again for your patient guidance!

2. Questions for General Evaluation

Reviewer’s Evaluation

Response and Revisions

Does the introduction provide sufficient background and include all relevant references?

Must be improved

We have made the appropriate changes and will respond to them in the point-by-point response letter.

Are all the cited references relevant to the research?

Yes

Is the research design appropriate?

Yes

Are the methods adequately described?

Yes

Are the results clearly presented?

Yes

Are the conclusions supported by the results?

Yes

3. Point-by-point response to Comments and Suggestions for Authors

Comments 1: Your paper is well written, with minor flaws in the English. It is a valuable report of your work.

Response 1: Thank you for recognizing our work, we will make changes based on the professional advice you give, which are explained below.

Comments 2: You report comparative tests of your results with other methods. Well done!

Response 2: Thank you for seeing how hard we work!

Comments 3: Your results are good and impressive.

Response 3: Your approval is our greatest encouragement!

Comments 4: A few papers have not been referred to. The recent papers [1–5] seem related and relevant to this work. You should defend - I think - why you excluded them in your comparison. For instance: why isn’t the survey by Huan et. al in Applied Sciences [1] used by you?

1.        Huang, L.; Liu, X.; Wang, X.; Li, J.; Tan, B. Deep Learning Methods in Image Matting: A Survey. Appl. Sci. 2023, 13, 1–22.

2.        Kong, Y.; Li, J.; Hu, L.; Li, X. Semi-Supervised Learning Matting Algorithm Based on Semantic Consistency of Trimaps. Appl. Sci. 2023, 13, 1–17.

3.        Yang, Y.; Gou, H.; Tan, M.; Feng, F.; Liang, Y.; Xiang, Y.; Wang, L.; Huang, H. Multi-Criterion Sampling Matting Algorithm via Gaussian Process. Biomimetics 2023, 8, 1–18.

4.        Zhang, X.; Wang, G.; Chen, C.; Dong, H.; Shao, M. Semi-Supervised Portrait Matting via the Collaboration of Teacher–Student Network and Adaptive Strategies. Electronics 2022, 11, 1–18.

5.        Zou, D.; Chen, X.; Cao, G.; Wang, X. Unsupervised Video Matting via Sparse and Low-Rank Representation. IEEE Trans. Pattern Anal. Mach. Intell. 2020, 42, 1501–1514.

Response 4: We agree. First of all we are very sorry that we have not been able to follow these references in time [1-4], the [5] reference we have cited in the article sorted as [8]. Our manuscript was modified and is marked in yellow in the revised document. Thank you very much for referencing such excellent references for us, we have read them carefully and cited references [1-4] in the following places:

1. Page 1

It is one of the key techniques in image processing and has a wide range of applications in practical scenarios, such as in image and video editing[1].

2. Page 2  

Kong et al. [10] proposed a semi-supervised deep learning matting algorithm based on semantic consistency of trimaps.

3.  Page 4   2.1. Image Matting

Yang et al.[22] develop a multi-criterion matting algorithm via Gaussian process, which searches for the optimal pixel pair by using the Gaussian process fitting model instead of solving the original pixel pair objective function.

4. Page 4   2.1. Image Matting

Zhang et al. [26] designed semi-supervised network to reduce complete dependency on labeled datasets.

Comments 5: Line 40, ‘ there have been a large amount algorithms.’ should read ‘many algorithms have been developed.’

Response 5: Thank you for pointing this out. We agree with this comment. We apologized for the lack of accuracy. We have revised this sentence in page. Our manuscript was modified and is marked in yellow in the revised document.

‘many algorithms have been developed. ‘ 

Comments 6:  Line 45, the shortcut ‘mattingV2(BGMv2)’ should be introduced after mentioning the name of its author.

The uninitiated reader is not aware yet what this shortcut is. You might in the beginning refer to the clarifying Tables 1 and 2, wherein all the shortcuts of methods are listed.

Response 6: We agree. We think your suggestion is more scientific and reasonable. We have changed this sentence in page 2. Our manuscript was modified and is marked in yellow in the revised document.

Subsequently, Lin et al. [7] proposed the background mattingV2(BGMv2), which focuses on high-resolution video matting with improved accuracy and efficiency but needs help with the auxiliary input and is still unsatisfactory in dealing with large dynamic video matting.

Comments 7: Line 97, ‘but better performance’ should read ‘and better performance’.

Response 7: Thank you for pointing this out. We agree with this comment. We have revised this sentence in page 3. Our manuscript was modified and is marked in yellow in the revised document.

‘and better performance’

Comments 8: Line 124, ‘the probability of indicating that the pixel is’ should read ‘the probability indicating that the pixel is’, or should read ‘the probability of indication that the pixel is’.

Response 8: We agree. Thank you for your careful review! We have revised this sentence in page 3. Our manuscript was modified and is marked in yellow in the revised document.

‘the probability of indication that the pixel is’

Comments 9:  Line 352, ‘Still, our method is easier to’ should read ‘Moreover, our method is easier to’.

Response 9:  We agree. We have revised this sentence in page 10. Our manuscript was modified and is marked in yellow in the revised document.

‘Moreover, our method is easier to’

Comments 10: Lines 363, 365, in the capture of the tables,  should have reference – for the quick reader - to the formulas (6) to (9). This could be done as ‘Objective evaluation metrics (6) to (9) of the four methods’.

Response 10: Thank you for pointing this out. We agree with this comment. We have changed these sentences in page 10. Our manuscript was modified and is marked in yellow in the revised document.

Table 1: Objective evaluation metrics formulas (6) to (9) of the four methods for image matting on the same dataset (bold indicates the best-ranked value).

Table 2: Objective evaluation metrics formulas (6) to (9) of the four methods for video matting on the same dataset (bold indicates the best-ranked value).

Comments 11: Line 371, Inconsistency: the name MORLIPO is not in the results Tables 1 and 2.

Response 11: We agree. We have revised the method names in Tables 1 and 2. Our manuscript was modified and is marked in yellow in the revised document.

Replace ours with MORLIPO.

4. Response to Comments on the Quality of English Language

Point 1: Well written, with minor flaws. See my points above: 5, 7, 8 and 9.

Response 1: Thank you very much for your recognition. We have revised items 5,7,8,9 in response to the comments and listed them in detail in the third point.

5. Additional clarifications

We would like to express our sincere appreciation to the editor and all the reviewers for their valuable comments and suggestions. We revised the paper accordingly. The major issues addressing these comments are explained below and are presented in the revised version of our manuscript. The modified parts are marked in the modified paper. We greatly appreciate your help and that of the reviewers to improve this paper. We hope that the revised manuscript is now suitable for publication.

Round 2

Reviewer 3 Report

Comments and Suggestions for Authors

Dear Authors,

I accept the changes you've made to the manuscript and therefore the research can be published.

All the best,